# The Molecular Biology of Prostate Cancer Stem Cells: From the Past to the Future

**DOI:** 10.3390/ijms24087482

**Published:** 2023-04-19

**Authors:** Yong Zhou, Tian Li, Man Jia, Rongyang Dai, Ronghao Wang

**Affiliations:** Department of Biochemistry and Molecular Biology, School of Basic Medical Sciences, Southwest Medical University, Luzhou 646000, China

**Keywords:** PCa, cancer stem cells, signaling pathway

## Abstract

Prostate cancer (PCa) continues to rank as the second leading cause of cancer-related mortality in western countries, despite the golden treatment using androgen deprivation therapy (ADT) or anti-androgen therapy. With decades of research, scientists have gradually realized that the existence of prostate cancer stem cells (PCSCs) successfully explains tumor recurrence, metastasis and therapeutic failure of PCa. Theoretically, eradication of this small population may improve the efficacy of current therapeutic approaches and prolong PCa survival. However, several characteristics of PCSCs make their diminishment extremely challenging: inherent resistance to anti-androgen and chemotherapy treatment, over-activation of the survival pathway, adaptation to tumor micro-environments, escape from immune attack and being easier to metastasize. For this end, a better understanding of PCSC biology at the molecular level will definitely inspire us to develop PCSC targeted approaches. In this review, we comprehensively summarize signaling pathways responsible for homeostatic regulation of PCSCs and discuss how to eliminate these fractional cells in clinical practice. Overall, this study deeply pinpoints PCSC biology at the molecular level and provides us some research perspectives.

## 1. Introduction

Prostate cancer (PCa) is the most common urological malignancy and brings great health threats to men. A recent global survey indicated that approximately 1.4 million new cases of and 375,000 deaths from PCa were documented in 2020 [1], which will display an increasing trend as PCa is an age-related disease. Prostate gland epithelial cells are mainly composed of two primary types of cells [2,3]: basal cells and luminal cells. Luminal epithelial cells are those well-differentiated androgen-dependent cells characterized by the high expression of androgen receptors (AR) and cytokeratins (CKs) [4,5,6], while basal epithelial cells tend to express a cluster of differentiation molecule 44 (CD44), keratin 5 (KRT5) and tumor protein 63 (P63) [4,7,8,9,10]. PCa shares biological similarity with luminal epithelial cells and is primarily driven by AR signaling [4,11,12]. PCa is dormant, and newly diagnosed cases only need active surveillance by periodically performed prostate specific antigen (PSA) testing, digital rectal examination (DRE) and biopsy [13,14].

It is worth noting that the 5 year survival rate of localized PCa is closed to 100%. However, PCa with bone or lymph node metastasis only has an approximately 30% 5 year survival rate, which is the main cause of PCa-related death [15]. Thanks to the pioneer study by Charles Huggins et al., androgen deprivation therapy (ADT) has been utilized as the golden management of advanced PCa [16]. Unfortunately, the duration of ADT only lasts 2–3 years and castration-resistant prostate cancer (CRPC) eventually develops [17,18]. At this stage, AR signaling is reactivated via several mechanisms and nicely responds to the castrated androgen level, supporting CRPC survival [19,20,21,22]. To further block AR signaling, the more powerful anti-androgen enzalutamide and adrenal androgen synthesis inhibitor abiraterone have been approved by the FDA to treat metastatic CRPC [23,24,25,26], with promising survival benefit. However, both of them only prolong CRPC patients’ survival for about 5 months.

There is a small population of prostate cancer cells existing in PCa, called prostate cancer stem cells (PCSCs), prostate progenitor cells or prostate initiating cells, contributing to PCa development [27]. Mounting evidence suggests that they play overarching roles in PCa initiation, progression and AR targeted therapy resistance [28,29], suggesting that eradication of this tiny population may be a promising strategy to fight against PCa. However, it is a big challenge to diminish such populations because they are inherently resistant to current treatments and extremely adaptive to the tumor’s micro-environment. Moreover, PCSCs evolve to activate survival signaling but suppress apoptosis signaling. For this end, a deep understanding of the molecular biology of PCSCs may give us some insights on how to eliminate these cells. This review will summarize the signaling pathways to this date related to the biology of PCSCs and discuss their potentially clinical applications.

## 2. Prostate Cancer Stem Cell

Cancer cell heterogeneity can result from the differentiation program driven by a subset of unique cancer cells called cancer stem cells (CSCs), which have self-renewal potential and can differentiate into various types of cells in a symmetrical or asymmetrical cell divided manner [30,31,32]. In 2005, with the successful isolation of CD44+/α2β1^hi^/CD133+ enriched cells from a human prostate tumor, Collins et al. found that this small population of PCa cells possessed self-renewal ability and were enabled to differentiate into non-clonal tumor cells [33]. Other studies also demonstrated that CD44+ and PSA^-/lo^ PCa cells shared similar properties with CD44+/α2β1^hi^/CD133+ enriched cells [34,35,36]. With decades of research, scientists have gradually recognized the existence and significance of these fractional cells in PCa using various biomarkers (Table 1) [37,38,39,40,41,42,43,44,45,46,47,48,49,50,51,52,53,54,55,56,57,58,59,60,61], which have stem cell characteristics and are then referred as PCSCs. Now, it is clear that PCSCs are involved in the initiation, progression and therapeutic resistance of PCa.

### 2.1. The Role of PCSCs in PCa Initiation

A number of literature studies have highlighted cancer stem cells as the origin of tumor initiation, including PCa. In the prostate gland, a small population of prostate stem cells (PSCs) undergo an oncogenic transformation to PCSCs when the external environment is unfavorable (Figure 1). Then, the omnipotent PCSCs give rise to various sub-types of prostate cancer cells, either AR-dependent or -independent, forming PCa heterogeneity. It seems that luminal-like PCa originates from progenitor cells in luminal epithelial cells. However, α2β1^hi^/CD133+ prostate stem cells were previously identified to be located in the basal layer [62]. Moreover, Witte et al. impressively found that it was the basal epithelial cells, but not luminal epithelial cells, growing to the prostate gland structure when they were subcutaneously implanted into nonobese diabetic scid gamma (NSG) mice [63]. Another study revealed that only basal epithelial cells with oncogenic induction led to prostatic intraepithelial neoplasia (PIN) and PCa in NSG mice [64]. All these findings together suggest that PCSCs in basal epithelial cells are the origin of PCa.

However, the origin of PCa in mice is different from that in humans. Studies by lineage tracing system elucidated that PSCs can be found in both the luminal and basal layers [65,66]. Further investigation illuminated that phosphatase and tensin homolog (PTEN) loss in either the basal layer or luminal layer led to PCa development. In the view of a CSCs-based hypothesis, PSCs in either the luminal layer or basal layer will transform to PCSCs when suffering from genetic alteration such as PTEN loss [67]. Although there is a little discrepancy between mouse PCa and human PCa regarding their origin (Figure 1), the concept of PCSCs as PCa origin has been widely acknowledged by many scientists: PSCs in the prostate epithelial cells undergo genomic instability to give rise to PCSCs when the environment is unfavorable, finally leading to PCa development.

### 2.2. The Role of PCSCs in PCa Progression and Anti-Androgen Resistance

PCa with 2–3 years ADT treatment will eventually develop to castration-resistant disease, a lethal stage owing to its metastatic potential. By bypassing this systemic androgen suppression, AR signaling hijacks several means in order to efficiently respond to the castrated androgen level. AR amplification, AR mutation and selective up-regulation of AR co-regulators are the more frequently observed biological processes in CRPC as compared with primary tumors [12,68,69,70].

Growing evidence also illustrates that ADT treatment expands the population of PCSCs [29,71,72], which have no response to androgenic sources. Continuous AR signaling inhibition of CRPC is fulfilled by the treatment of FDA approved anti-androgen enzalutamide [23,26], which competes androgen to bind AR and suppresses AR nuclear imports [73]. However, it only benefits CRPC by extending to a five month survival [74]. One popular mechanism responsible for the acquired enzalutamide resistance is the selective expression of truncated AR variants. For example, ARv7, which lacks an androgen responsive domain, is a crucially causal factor controlling enzalutamide resistance [75,76,77]. Another important mechanism responsible for anti-androgen resistance is lineage plasticity [78], which is possibly driven by PCSCs. Accumulative evidence suggests that AR-targeted therapies including ADT and anti-androgen treatment considerably enrich the population of PCSCs by facilitating the transition of AR-dependent PCa cells to AR-independent cells with stem-cell-like states or multi-lineage stages [79,80]. PCSCs then re-differentiate into AR-therapy-resistant sub-clones such as neuroendocrine PCa cells or other lineage sub-clones inherently resistant to enzalutamide treatment. In this regard, PCSCs play critical roles in PCa progression and anti-androgen resistance.

## 3. Signaling Pathways Maintain Prostate Cancer Stem Cell Population

As PCSCs play critical roles in PCa development and they are currently undruggable, understanding their biology is the priority if we want to minimize this small population. Therefore, we turn our focus to the signaling pathways that are causally linked to the survival and expansion of the PCSC population (Figure 2). In our opinion, this review of PCSC biology at the molecular level would interest a wide range of readers and inspire them to develop PCSC targeted therapy.

### 3.1. Janus Kinase-Signal Transducer and Activator of Transcription (JAK-STAT) Signaling Pathway in PCSCs

The JAK-STAT pathway is implicated into various processes such as cell proliferation, anti-apoptosis and immune response [81,82]. The direct interaction of ligands (cytokines and interleukins) with the cell surface receptor leads to the trans-phosphorylation of JAK, which is recognized by the STAT protein. Subsequently, JAK phosphorylates the specific tyrosine residue of STAT and leads to its activation. Phosphorylated STAT dissociates from JAK and enters the nucleus as dimer to output its transcriptional activity [83,84].

STAT3, a member of the STAT family, has been illustrated as one of the fundamental driving forces to control the occurrence and development of various tumors [85,86,87,88]. In particular, STAT3 signaling is prone to be activated in CSC sub-populations. The maintenance of CSCs by STAT3 is probably attributable to its transcriptional regulation on CD44, CD133 and epithelial–mesenchymal transition (EMT) related genes [89,90]. Reciprocally, nuclear CD44 can complex with STAT3 and enhance its transcriptional activity [91,92]. In PCa, gene expression signature analysis identified that JAK1-STAT3 signaling was highly enriched in the CD133+/α2β1 integrin^high^ stem-like PCa sub-population [93]. It also demonstrated that STAT3 activation was a prerequisite for the survival of this type of PCSC [94]. Interestingly, the study also showed that ADT-driven AR inhibition could increase interleukin 6 (IL-6) levels, which triggers STAT3 activation by binding to glycoprotein 130 (gp130) receptors and leads to the subsequent expansion of the PCSC population, further supporting the critical role of STAT3 signaling in PCSC evolution [90,95]. In addition to IL6/gp130, STAT3 activation can be fulfilled by other factors. The literature reveals that monoamine oxidative A (MAOA), HOX transcript antisense RNA (HOTAIR), interleukin 10 (IL-10) and bacterial lipopolysaccharide (LPS) enable an increase in PCSC population via driving STAT3 signaling [96,97,98].

Now, new evidence illustrates that STAT1 can cooperate with STAT3 to promote the transition of AR-dependent PCa to stem-like cells with multilineage potential in an enzalutamide treated PCa model [28]. In this model, the authors found that SOX2 physically binds to the promoter region of *JAK1*, *STAT1* gene locus and increases their expression. Activated JAK1-STAT1 signaling regulates the expression of critical genes indispensable for lineage plasticity transition, conferring anti-androgen resistance. Indeed, genetic and pharmaceutical inactivation of JAK1 can reverse the acquisition of lineage plasticity mediated enzalutamide resistance of PCa cells and human derived organoids.

Together, both STAT1 and STAT3 are central players during the process of anti-androgen-induced PCSC evolution, suggesting that combined therapy using their specific inhibitors would be a promising strategy to diminish PCSC populations and to overcome anti-androgen resistance.

### 3.2. AR Signaling in PCSCs

As one member of nuclear receptor, AR and NR3C4 have been widely acknowledged as the driving factor determining PCa initiation and progression. Once responding to androgenic hormones, AR imports to the nucleus and serves as a homo-dimer or hetero-dimer to regulate the expression of a variety of genes, supporting PCa survival [99]. The significance of AR in PCa development leads ADT and anti-androgen as the standardized therapies against this type of disease. However, studies have revealed that ADT and anti-androgen treatment result in the back-differentiation of some AR-positive PCa cells into PCSCs, which are inherently resistant to AR targeted therapies due to their negative AR characteristics. In a murine PCa model, the inhibition of AR with flutamide-activated IL-6/STAT3 signaling awakened PCSC growth, consistent with the aforementioned demonstration of IL-6/STAT3 signaling for the maintenance of PCSCs population [90]. Indeed, combined therapy of the IL-6 receptor fusion protein IL-6^RFP^ and flutamide could synergistically suppress murine PCa growth by blocking ADT-induced PCSC expansion. Accordingly, another literature study also suggested that NANOG, a critical stemness associated transcription factor, reprograms LNcaP cells to castration resistance via occupying the chromatin binding sites of AR [100]. Collectively, these data illuminate that AR signaling exerts an inhibitory effect on PCSC expansion. Contradictory to this notion, the results from Cai et al.’s study demonstrated that synergistic activation of kirsten rat sarcoma virus G12D (Kras G12D) and AR in murine prostate cells resulted in aggressive PCa by expanding the PCSC population [101]. According to their data, Kras cooperates with AR to increase the expression level of the enhancer of zeste homolog 2 (Ezh2), which acts as one component of the polycomb repressive complex 2 (PRC2) complex to suppress the expression of differentiation-associated genes. For us, although AR/Kras derived PCa tumors have high levels of Ezh2, AR status in Ezh2-enriched PCSCs at the single cell level is still questionable. Overall, growing evidence suggests that PCSCs rarely express AR and AR, at least alone, suppresses PCSC survival.

### 3.3. Hedgehog (Hh) Signaling Pathway in PCSCs

Hedgehog signaling pathway activated by Hh ligands, including Sonic Hh (SHh), Indian Hh (IHh) and Desert Hh (DHh), is implicated into all aspects of cancer evolution [102]. The binding of Hh ligands to the twelve-pass trans-membrane protein receptor Patched1 (PTCH1) relieves the inactive Smoothened (SMO), a G protein-coupled receptor [103,104]. The active SMO, together with β-Arrestin (Arrb2) and kinesin family member 3A (Kif3a), promotes the trans-activation of full length glioma-associated oncogene homolog 1-3 (GLI1-3), regulating Hh targeting genes involved in the regulation of multiple aspects of tumorigenesis, including CSC homeostasis [105,106,107]. Indeed, literature by Han-Hsin Chang et al. demonstrated that aberrant activation of Hh signaling led to the transformation of normal prostate basal/stem cells into PCSCs, which further differentiated into metastatic cancer cells [108]. Results from another report revealed that intraprostatic delivery of Hh ligand to mice resulted in the development of aggressive prostate cancer [109]. Interestingly, the active Hh ligand was localized to the area of p63 positive cells, implying that Hh-induced PCa aggressiveness is attributable to the enrichment of PCSCs. In addition, genistein suppresses the tumorsphere formation of prostate cancer cells by acting as an Hh signaling inhibitor [110]. Together, all this evidence strongly reinforces the significant role of the Hh signaling pathway in PCSC evolution.

### 3.4. Wnt Signaling Pathway in PCSCs

Mounting evidence supports that the Wnt signaling pathway is indispensable for CSC homeostasis [111,112]. In PCa, the binding of the Wnt ligand with its receptor Frizzled and lipoprotein receptor-related protein (LRP)5/6 results in the stabilization and nuclear translocation of β-catenin, which acts as a cofactor of the T-cell factor/lymphoid enhancer factor (TCF/LEF) transcription factor family to regulate PCSC-related genes such as CD44 [113]. The literature demonstrates that β-catenin and its downstream targeting genes, such as axis inhibition protein 2 (Axin 2) and cyclin D, were highly enriched in human telomerase reverse transcriptase (hTERT) mediated PCSC traits [114]. Down-regulation of β-catenin by siRNAs remarkably attenuated hTERT mediated CSC features of prostate cancer cells. The critical role of Wnt signaling in maintaining the biology of PCSCs has also been strengthened by several literature studies, suggesting that the activation of Wnt signaling by PHD finger protein 21B (PHF21B), endothelial cell-specific molecule 1 (ESM1) or anaplastic lymphoma kinase (ALK) promotes PCSC phenotype [115,116,117]. The negative correlation of AR with β-catenin in primary PCa is shifted to the positive correlation in CRPC, implying that Wnt induced PCSC populations may contribute to castration resistance [118]. Indeed, the overexpression of active β-catenin results in resistance to castration, and the inhibition of Wnt signaling could overcome this phenomenon by reducing PCSC populations. Interestingly, a recent Science article demonstrated that PCa can be classified into four sub-types according to the molecular signature: AR-dependent PCa, cancer stem-like PCa, neuroendocrine PCa and Wnt-dependent PCa, highlighting the heterogeneous role of Wnt signaling in driving PCa progression, in addition to its definite role in the regulation of PCSC biology [119]. Indeed, β-catenin has been documented to activate AR signaling by forming a complex with AR. In conclusion, the Wnt signaling pathway is essential for the stemness of prostate cancer cells, even though its precise role in PCa development requires deep investigation.

### 3.5. Notch Signaling Pathway

Notch is a genetically conservative signaling transduction pathway implicated into various biological processes [120,121]. Interactions of Notch receptors (Notch1-4) with the specific ligands (Delta-like 1, Delta-like 3, Delta-like 4, Jagged-1 and Jagged-2) between adjacent cells triggers the proteolytic cleavage of the receptor and results in the release of the Notch intracellular domain (NICD), which trans-locates into the nucleus and guides a transcriptional switch of the CSL [CBF1/Su(H)/Lag-1] complex by releasing corepressors and recruiting coactivators. Cytoplasm-retained NICD is also functional via cross-talks with other signaling pathways such as protein kinase B (AKT), mammalian target of rapamycin complex 2 (mTORC2) and nuclear factor kappa-light-chain-enhancer of activated B cells (NF-κB). Now, it is clear that abnormal activation of Notch signaling observed in PCa is causally linked to its initiation, angiogenesis, metastasis, drug resistance and CSC enrichment [122,123,124]. Numb^-/low^ CRPC sub-population with stem-like cell property prefers to express Notch targeting genes as compared with Num^high^ counterparts [125]. A recent study has shown that bone marrow mesenchymal stem cells (BMSCs) promote the stemness of prostate cancer cells via activating the Jagged1/Notch1 pathway [126]. Recently, Kufe et al. illustrated that MUC1-C can drive the expression of several BRG1/Brahma-associated factor (BAF) complex components in an E2F1-dependent manner. As feedback, BAF forms a nuclear complex with MUC-C1 to induce Notch1 expression and to promote the stemness of PCa [127]. Collectively, these data suggest that Notch signaling is a key player to determine PCa stem-like traits.

Given the fact that PCSC population enrichment is always considered as one critical resistant mechanism to anti-androgen or docetaxel treatment in PCa, targeting Notch signaling to eliminate PCSCs may increase the anti-androgen/docetaxel efficacy. Indeed, PF-03084014, a γ-secretase inhibitor (GSI) that prevents the cleavage of Notch receptors, can reduce the stemness characteristics of docetaxel-resistant or enzalutamide-resistant CRPC cells and restore drug sensitivity [128,129]. Taken together, Notch signaling is over-activated in PCSCs and its inhibition is one potential strategy to minimize PCSC populations.

### 3.6. NF-κB Signaling Pathway in PCSCs

The nuclear transcription factor NF-κB family has five members (p65, RELB, c-REL, p50 and p52) and has been documented to control cancer cell survival, metastasis and CSC homeostasis [106,130,131]. Canonical NF-κB signaling is initiated by the activation of the IκB kinase (IKK) complex, which leads to the phosphorylation and subsequent degradation of IκB. Without the mask of IκB, NF-κB members are released and translocated from the cytoplasm to the nucleus as either homo-dimers or hetero-dimers, enabling the transcription of targeted genes via specifically binding to its responsive DNA element [132,133]. In addition, NF-κB signaling can be non-canonically activated in the form of p52/RELB dimer by other receptors such as B-cell activation factor (BAFFR), CD40, receptor activator for nuclear factor kappa B (RANK) and lymphtoxin b-receptor (LTbR) [134,135].

The role of NF-κB in the homeostatic regulation of PCSCs was recognized by Rajasekhar et al., illustrating that isolated human tumor-initiating cells (TICs) with stem-cell-like properties exhibited active NF-κB activity [136]. Significantly, the attenuation of NF-κB signaling by its specific inhibitors, such as 481,407 compound, parthenolide and celastrol, suppressed the secondary sphere formation of TICs, supporting that NF-κB activation is a prerequisite for PCSC expansion. In line with this finding, C-X-C motif chemokine ligand 12 (CXCL12) gamma, with its receptor C-X-C motif chemokine receptor 4 (CXCR4), enhanced PCSC phenotype by activating NF-κB signaling [137]. Now, it is clear that the stemness related to NF-κB signaling is at least partially attributable to its direct regulation on SOX9, one critical factor consistently increased in stem-like prostate cancer cells [138]. In the future, exploration of NF-κB downstream effectors in regard to the regulation of PCSC homeostasis deserves scientists’ considerable efforts.

### 3.7. Phosphoinositide 3-Kinase/Protein Kinase B (PI3K/AKT) Signaling Pathway in PCSCs

PI3K/AKT is a highly conserved signaling pathway, and its disordered regulation is frequently observed in cancer development [139,140]. Usually, ligand bound membrane receptors such as G protein-coupled receptors (GPCR) or tyrosine kinase receptor (RTK) mediated PI3K activation results in the conversion of phosphatidylinositol 4,5-bisphosphate (PIP2) to phosphatidylinositol 3,4,5-triphosphate (PIP3), which binds to AKT at the plasma membrane and leads to AKT phosphorylation by phosphoinositide-dependent kinase 1 (PDK1). Phosphorylated AKT is activated and exerts its biological functions via regulating a variety of downstream molecules such as tuberous sclerosis protein 2 (TSC2), forkhead box transcription factors of the class O (FOXO) and glycogen synthase kinase 3b (GSK3b) [141,142,143,144].

Aberrant activation of PI3K/AKT signaling by PTEN loss is frequently observed in 30% primary PCa and 60% CRPC [145]. The contribution of this signaling to PCSC expansion was recognized as early as 2009, when a literature study demonstrated that prostate cancer progenitors (PCaPs) cultured in sphere-forming conditions preferred to activate PI3K/AKT signaling by overexpressing PI3K p110a/b protein levels [146]. Mechanistically, the enrichment of the PCSC population was at least partially due to AKT-mediated nuclear export and degradation of FOXO3a, one critical transcription factor regulating the stemness of CSCs. Additional evidence suggested that the inhibition of PI3K/AKT signaling by LY294002 and NVP-BEZ235 indeed noticeably suppressed the survival and sphere forming ability of PCa cells [147,148]. In agreement with this finding, L Chang and his colleagues also elucidated that the PI3K/AKT regulated stemness phenotype was causally associated with the radio-resistance of PCa [149]. The significant and complex role of PI3K/AKT signaling in PCa prompted clinical trials of its inhibitors as a therapeutic strategy for metastatic CRPC, although they finally failed.

New evidence suggests that the strong inhibition of AR by the second anti-androgen selects activating PI3K/AKT signaling to develop anti-androgen resistant CRPC [150]. We thus believe combined therapy targeting AR and PI3K/AKT signaling is a fascinating strategy to treat metastatic CRPC, and is indeed intensively tested in clinical trials.

### 3.8. Hippo Signaling Pathway

The Hippo signaling pathway acts as a crucial regulator during cell growth and organ development and is controlled by a kinase cascade [151,152]. In the active status, a mammalian STE20-like protein kinase 1/2 (MST1/2), G protein coupled receptor (GPCR), or mitogen-activated protein kinase kinase kinase kinase (MAP4K) family member acts as the upstream molecule to phosphorylate tumor suppressor kinase LATS1/2 (large tumor suppressor kinase 1/2), which subsequently phosphorylates Yes-associated protein (YAP) and transcriptional co-activator with PDZ-binding motif (TAZ), leading to their cytoplasmic retention and degradation [153]. When Hippo signaling is silent, YAP/TAZ is dephosphorylated and translocated into the nucleus, initiating gene transcription via binding to the TEAD family of transcription factors [154,155,156].

Activation of YAP/TAZ caused by either mutation on Hippo pathway upstream molecules or the cellular regulatory network promotes tumorigenesis and cancer stem-like phenotype [157,158]. The role of the Hippo pathway in PCSC biology has been recognized by this study, which demonstrated that the inhibition of phosphodiesterase 5 (PDE5) by its specific inhibitor vardenafil attenuated PCSC phenotype by activating cGMP-dependent protein kinases (PKGs), which triggered the Hippo signaling cascade by directly phosphorylating MST1/2, thereby degrading TAZ to achieve the purpose of downregulating the stemness of PCSCs [159]. In addition, CD44, a popular biomarker of PCSCs, promoted the invasion and metastasis of docetaxel-resistant PCa cells by inducing the expression of YAP [160]. Together, the Hippo inactivation mediated nuclear accumulation of YAP/TAZ accentuated PCSC phenotype and configuration of YAP/TAZ transcriptional complex or the upstream regulators may become the research direction in this field.

### 3.9. The Activator Protein-1 (AP1) Transcription Factor

AP-1 protein is a transcription factor with a basic leucine zipper domain and acts as a homo-dimer or hetero-dimer to regulate gene expression, participating in various cellular processes including proliferation and differentiation [161,162]. A very recent literature study published in Science identified that AP1 family member FOS-like 1 (FOSL1) ranks the top key transcription factor to determine the program of CRPC-CSC [119]. In this study, the authors utilized the ATAC-seq from 40 metastatic prostate cancer models (twenty-two patient-derived organoids, six patient-derived xenografts, seven cell lines and five neuroendocrine PCa patients) as inputs to perform clustering consensus analysis based on the top variable chromatin accessible peaks, leading to the identification of four molecular sub-types of CRPC: AR-dependent PCa (CRPC-AR), cancer stem-like PCa (CRPC-SCL), neuroendocrine PCa (CRPC-NE) and Wnt-dependent PCa (CRPC-Wnt). CRPC-SCL, accounting for an estimated 20% of CRPC patients, displays a highly prostate basal stem cell signature compared with other three sub-types and experiences shorter time on AR signaling inhibitor (ARSI) treatment than that of its CRPC-AR counterparts. By integrating ATAC-seq and RNA-seq data, they constructed a regulatory network and finally identified AP-1 family protein FOSL1 as the master regulator for CRPC-SCL. Mechanistically, FOSL1 cooperated with TEAD, YAP and TAZ to control the transcriptional program of CRPC-SCL, which was also confirmed by the overlapped ChIP-seq peaks of these four proteins in SCL models. Indeed, individual depletion of FOSL1, YAP, TAZ and TEAD evidently suppressed the cell growth of SCL models, but had a negligible effect on AR-dependent PCa models. Similar results were also obtained when AP1 inhibitor T-5224 and YAP/TAZ inhibitor verteporfin were tested. In summary, the collective data illustrate that FOSL1 plays as a pioneer factor or a master regulator to control the PCSC transcriptional program, and it will be fascinating to see the clinical outcomes of the AP1 inhibitor in PCa management.

### 3.10. Others

In addition to the aforementioned signaling pathways, other molecules have been recently reported to influence PCa stemness. Bromodomain adjacent to zinc finger domain protein 2A (BAZ2A), a bromodomain (BRD)-containing protein, recognized histone acetyltransferase p300 (EP300)-mediated H3K14ac marks and suppressed the transcription of differentiation-related genes such as aldehyde oxidase 1 (AOX1), DNA damage-binding protein 1 (DDB1), beta-mannosidase (MANB2) and dystrophia myotonica protein kinase (DMPK), maintaining the stemness stage of PCa [163]. Either BAZ2A mutation or pharmacological inactivation by BAZ2A inhibitor GSK2801 or BAZ2-ICR significantly impaired PCa stem-like features. Although non-coding RNAs, including miRNAs and long non-coding RNAs, have also been reported to be involved in the maintenance of PCSCs, they do not function independently and are therefore excluded from this review.

Glucocorticoid receptor (GR), another steroid receptor member, is also implicated into PCSCs development. Very interestingly, chronic enzalutamide treatment leads to GR upregulation, which replaces AR to transcriptionally regulate AR targeting genes for the survival of enzalutamide-resistant PCa, a new mechanism for PCa cells to escape AR inhibition [164]. Just like the role of AR in PCSCs, GR is also a negative player in PCSC development. A study from Maitland et al. suggested that GR could increase the expression levels of miR-99a and miR-100, which would disrupt the stem cell phenotype of PCa by directly targeting SMARCA5 (an SWI/SNF-related matrix-associated actin-dependent regulator of chromatin subfamily A member 5) and SMARCD1 (an SWI/SNF-related matrix-associated actin-dependent regulator of chromatin subfamily D member 1) [165]. Together, these results inspire us to combine GR inhibitor with PCSC targeted therapy for the management of enzalutamide-resistant PCa.

## 4. Potential Clinical Applications of Targeting PCSCs

As discussed, Hedgehog, Wnt, Notch, Hippo, PI3K/AKT, AP1, NF-κB and JAK-STAT signaling pathways control the survival and homeostasis of PCSCs. To eradicate this fractional population, specific inhibitors towards these signaling pathways have been tested or are being tested in PCa patients/models (Table 2) [128,163,166,167,168,169,170,171,172,173,174,175,176,177,178,179,180,181,182,183,184,185,186,187,188,189,190,191,192,193,194,195,196,197,198,199]. For example, gamma-secretase inhibitor RO4929097, for the shutdown of NCID production, was used in combination with bicalutamide to treat PCa patients in 2010 (NCT01200810) [29]. Another clinical trial (NCT01695473) using PI3K/AKT inhibitor BKM120 as a neoadjuvant agent was performed in high risk localized PCa patients [200]. Although these two clinical trials failed, the strategy opens a new window for PCa management by diminishing PCSCs. Additionally, other pathway inhibitors have also been tested in preclinical PCa models: Hedgehog pathway inhibitor sonidegib has been shown to suppress the transcription of stemness-related genes to slow PCa growth [201]; Wnt signaling inhibitors such as 3289-8625, Foxy-5 and OMP-54F28 remarkably hinder PCa growth [178,202,203]; AP1 inhibitor T-5224 and YAP/TAZ inhibitor verteporfin specifically suppress the survival of PCSCs [119]; and JAK-STAT signaling pathway inhibitor ruxolitinib or the combination of fludarabine and niclosamide all show a great inhibitory effect on enzalutamide-resistant PCa models [28].

Immunotherapy is another therapeutic strategy to treat metastatic CRPC patients [204,205]. For instance, prostate specific membrane antigen (PSMA) based CART cells (chimeric antigen receptor T cells) and immune checkpoint inhibitors have been investigated in preclinical PCa models and show beneficial outcomes [206]. Owing to the significance of PCSCs in PCa tumor recurrence, anti-androgen resistance and tumor metastasis, scientists are looking forward to see PCSC-based immunotherapies. In 2020, Chan’s group utilized two immunogenic peptides derived from CD44 and epithelial cell adhesion molecule (EpCAM) to educate the isolated dendritic cells, followed by co-culture with cytokine-induced killer cells (CIKs). Interestingly, these adaptive CIKs displayed a tremendous inhibition on PCSCs in vitro and in vivo [207].

Although current therapies against PCSCs are not applicable in clinical management owing to various reasons such as drug toxicity, drug stability in the human body and delivery systems, it is still charming for scientists and clinicians to continuously explore in this way.

## 5. Conclusions and Future Perspectives

To summarize, we conclude that Hedgehog, Wnt, Notch, Hippo, PI3K/AKT, AP1, NF-κB and JAK-STAT signaling pathways have positive contributions to the evolution and development of PCSC populations, while AR works negatively. Now, scientists are continuously making efforts to identify other pathways/molecules necessary for PCSC homeostasis. To explore other potential pathways related to PCSCs, we performed Kyoto Encyclopedia of Genes and Genomes (KEGG) pathway analysis on the differential expressed genes (DEGs) between CRPC-AR and CRPC-SCL sub-populations classified by consensus clustering analysis on the SU2C CRPC dataset. Analysis revealed that neuroactive ligand–receptor interactions, MAPK signaling pathway, cytokine–cytokine receptor interactions, calcium signaling pathway and cAMP signaling pathway have potentials to regulate PCSC biology (Figure 3), which requires our experimental validations.

The PCa micro-environment at least consists of fibroblasts, endothelial cells and immune cells, providing growth support for cancer epithelial cells [208,209,210]. PCSCs are believed to reside in special micro-environments called niches [211], which undoubtedly has a great impact on PCSC survival and homeostasis. To end this, we performed immune cell infiltration analysis, and the result showed that the population of micro-environmental components is considerably different between CRPC-AR and CRPC-SCL, implying that the survival and expansion of PCSCs are also tightly regulated by the surrounding micro-environment. Again, tremendous efforts on this direction are required.

## Figures and Tables

**Figure 1 ijms-24-07482-f001:**
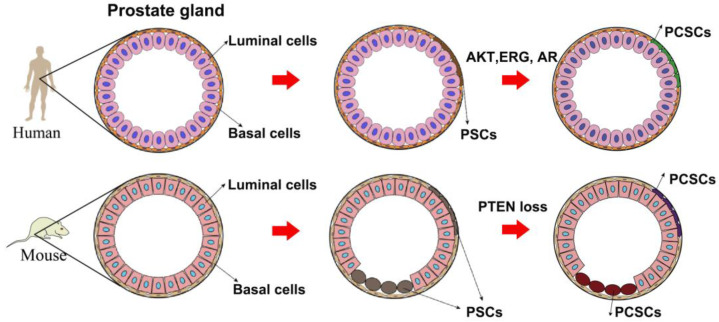
A model of the origin of prostate cancer stem cells.

**Figure 2 ijms-24-07482-f002:**
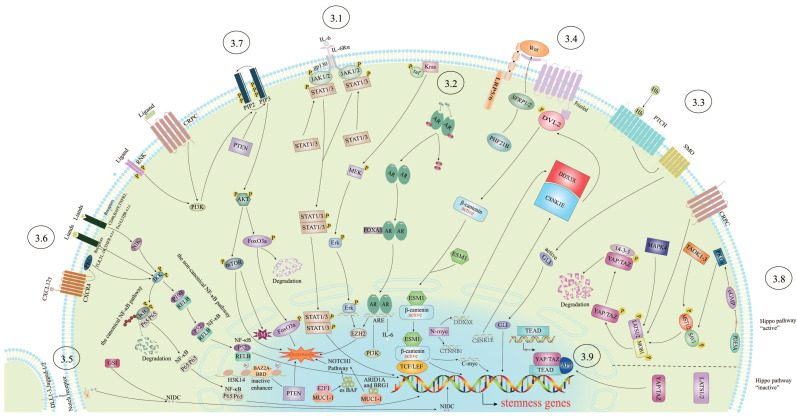
A schematic depiction of signaling pathways involved in PCSC biology.

**Figure 3 ijms-24-07482-f003:**
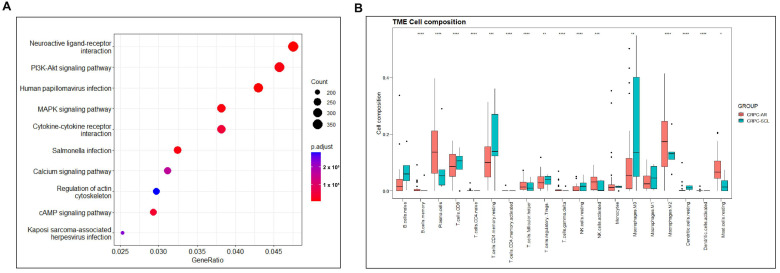
KEGG pathways analysis and immune infiltration analysis of the DEGs between CRPC-AR and CRPC-SCL. (**A**) KEGG pathway analysis of the DEGs between CRPC-AR and CRPC-SCL. (**B**) The differential immune cell infiltration between CRPC-AR and CRPC-SCL. * *p* < 0.5; ** *p* < 0.01; *** *p* < 0.001; **** *p* < 0.0001.

**Table 1 ijms-24-07482-t001:** Biomarkers for prostate cancer stem cells.

Marker	Function	Reference
CD44	CD44 is a glycoprotein involved in cell migration, adhesion and signal transduction	Kalantari E, et al., 2017 [37]
CD133	CD133 is a transmembrane glycoprotein involved in cell self-renewal, differentiation and tumor invasion	Kalantari E, et al., 2017 [37]
CD117	CD117 is a transmembrane glycoprotein involved in cell self-renewal, differentiation and tumor invasion	Harris KS, et al., 2021 [38]
α2β1	α2β1 is associated with tumor invasion and proliferation	Aldahish A, et al., 2019 [39] Naci D, et al., 2015 [40]
EpCAM	EpCAM is a calcium independent adhesion molecule between epithelial cells involved in the process of epithelial cell carcinogenesis	Witte KE, et al., 2021 [41]
SOX2	SOX2 plays an important role in maintaining stem cell pluripotency and self-renewal	Lee Y, et al., 2021 [42] Vaddi PK, et al., 2019 [43]
EZH2	EZH2 is closely related to cell migration and invasion, tumor development and stem cell self-renewal	Huang J, et al., 2021 [44]
CXCR4	CXCR4 is a specific receptor involved in physiological mechanisms such as HIV-1, hematopoiesis, embryonic development and tumor migration	Li Y, et al., 2019 [45] Chatterjee S, et al., 2014 [46]
TRA-1-60	A cell surface epitope of human embryonic, embryonal germline and teratocarcinoma stem cells	Schafer C, et al., 2020 [47]
CD151	CD151 is associated with tumor initiation, metastasis, and angiogenesis	Wong AH, et al., 2020 [48]
OCT-3/4	OCT-3/4 is an essential transcription factor that maintains the multidirectional differentiation potential of embryonic stem cells and primordial germ cells	Wang X, et al., 2021 [49] Fujimura T, et al., 2014 [50]
Smo	Smo is a transmembrane protein that mediates Hedgehog signaling to the intracellular compartment	Lou H, et al., 2020 [51]
Nanog	Nanog is a transcription factor with an important role in stem cell self-renewal and maintenance of pluripotency	Verma S, et al., 2020 [52] Liu C, et al., 2020 [53]
Bmi-1	Bmi-1 is associated with maintenance of self-renewal of prostate stem cells and inhibition of PTEN in PCa	Li Y, et al., 2017 [54] Yoo YA, et al., 2016 [55]
TWIST	TWIST is a transcription factor with a helix-loop-helix structure and associated with tumor invasion and metastasis	Lee Y, et al., 2021 [42]
CD24	CD24 is a cell adhesion molecule involved in the regulation of B-cell proliferation and maturation	Costa CD, et al.,2019 [56]
CD166	CD166 is a leukocyte adhesion factor associated with cell adhesion and tumor metastasis	Wei GJ, et al., 2019 [57] van Lersner A, et al., 2019 [58]
CD49b	CD49b is also called integrin α2, a cell surface receptor associated with adhesion and lymphocyte activation	Bansal N, et al., 2016 [59] Erb HHH, et al., 2018 [60]
ABCG-2	ABCG-2 contributes to the resistance to chemotherapeutic drugs	Kim WT, et al., 2017 [61]

**Table 2 ijms-24-07482-t002:** Prostate cancer stem cells related inhibitors.

Signaling Pathway	Name of Inhibitor	Target	Reference
Notch signaling pathway	PF-03084014	γ-secretase	Cui D, et al., 2015 [128]
	RO4929097	γ-secretase	Du Z, et al., 2021 [166]
	DAPT(GSI-IX)	Notch-1	Cui J, et al., 2018 [167]
	SOX8/RO04929097	Notch-4	Du Z, et al., 2022 [168]
Hedgehog signaling pathway	GDC-0449	SMO	Tong W, et al., 2018 [169]
	GANT-61	GLI	Tong W, et al., 2018 [169]
	vismodegib	SMO	Leao R, et al., 2017 [170]
	itraconazole	SMO	Leao R, et al., 2017 [170]
	sonidegib (LDE-225)	SMO	Burness CB, et al., 2015 [171]
Wnt signaling pathway	LGK974	Porcupine	Ma F, et al., 2016 [172]
	DDK1	LRP5	Browne AJ, et al., 2016 [173]
	DKK3	Frizzled/LRP5/6 complex	Bhattacharyya S, et al., 2019 [174]
	OMP-54F28 (Ipafricept)	Wnt family	Katoh M, et al., 2017 [175]
	OMP-18R5 (Vantictumab)	Frizzled1,2,5,7,8	Katoh M, et al., 2017 [175]
	SFRP	Wnt ligands	Cruz-Hernandez CD, et al., 2020 [176]
	ETC-159 (ETC-1922159)	Porcupine	Katoh M, et al., 2017 [175]
	ICG001	β-actin	Kelsey R, et al., 2017 [177]
	3289-8625	DVL	Schneider JA, et al., 2018 [178]
STAT3 signaling pathway	Napabucasin (BBI608)	STAT3 gene transcription	Zhang Y, et al., 2016 [179]
	Alantolactone (ALT)	STAT3(Tyr705)	Babaei G, et al., 2020 [180]
	EC-70124	STAT3(Tyr705)	Civenni G, et al., 2016 [181]
	ruxolitinib	JAK	Han IH, et al., 2020 [182]
PI3K/AKT signaling pathway	HIF1α	mTOR	Marhold M, et al., 2015 [183]
	NVP-BEZ235	PI3K and mTOR	Liu G, et al.,2020 [184]
	GSK2636771	P110β/γ	Sarker D, et al., 2021 [185]
	LY3023414 AZD8186	P110α/β	Choudhury AD, et al., 2022 [186] Hancox U, et al., 2015 [187]
	Apitolisib(GDC-0980), buparlisib	pan-PI3K	Zhou Y, et al., 2015 [188] Armstrong AJ, et al., 2017 [189]
	Ipatasertib MK2206 AZD5363	pan-AKT	Armstrong AJ, et al., 2017 [189] Sweeney C, et al., 2021 [190]
	Everolimus Temsirolimus Ridaforolimus AZD2014, MLN0128	mTOR	Eule CJ, et al., 2023 [191] Kruczek K, et al., 2013 [192] Meulenbeld HJ, et al., 2013 [193] Li S, et al., 2021 [194] Graham L, et al., 2018 [195]
NF-κB signaling pathway	EC-70124	IκB phosphorylation	Civenni G, et al., 2016 [181]
	Eupatilin	NF-κB	Serttas R, et al., 2021 [196]
	JSH-23	NF-κB	Chang KS, et al., 2019 [197]
Hippo signaling pathway	STK3/4	YAP1;TAZ	Schirmer AU, et al., 2022 [198]
	verteporfin	YAP;YAP/TEAD complex	Li Q, et al., 2021 [199]
Others	GSK2801	BAZ2A	Pena-Hernandez R, et al., 2021 [163]
	BAZ2-ICR	BAZ2A	

## Data Availability

The corresponding author will provide the raw materials if there is a request.

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
