# Peer review of "The Molecular Biology of Prostate Cancer Stem Cells: From the Past to the Future"

_ijms, 2023, doi:10.3390/ijms24087482_

Round 1

Reviewer 1 Report

Dear authors,

After reviewing your review article, I cannot actually understand the main rationale of thie study. The authors describe the purpose of this study in the abstract but not in the main text. Second, the authors describe that this study aims to summarize the molecular mechanism for androgen escape. However, the authors did not mention the main purpose you intend to summarize the molecular mechanism. Is there any challenge state in treating PCSCs in the clinics? The authors need to clearly mention the biggest challenges in this field.

Third, several references the authors cited in this study is too old. For review article, citing references older than 10 years is not recommended. Also, the style of the reference citing is not unified. I suggest the authors can base on the previous review (such as 10.3390/cancers11040434, and 10.3389/fonc.2022.935715) and update the latest finding in this review.

Forth, microRNAs and glucocorticoid receptor signaling are also play critical role in the stemness and androgen escape in PCa. I suggest the authors discuss them.

Author Response

Reviewer #1 (Remarks to the Author):

Q1. After reviewing your review article, I cannot actually understand the main rationale of thie study. The authors describe the purpose of this study in the abstract but not in the main text.

Ans: We appreciate your comment. As PCSCs play critical roles in PCa development and they are currently undruggable, understanding of their biology is the priority if we want to minimize this small population.Therefore, we turn our focus on the signaling pathways which are casually linked to the survival and expansion of PCSCs population. In our opinion, this review of PCSCs biology at the molecular level would interest a wide range of readers and inspire them to develop PCSCs targeted therapy. You can reach the rationale of this review in the abstract, paragraph 55-60, paragraph 128-133. Thanks

Q2. Second, the authors describe that this study aims to summarize the molecular mechanism for androgen escape. However, the authors did not mention the main purpose you intend to summarize the molecular mechanism. Is there any challenge state in treating PCSCs in the clinics? The authors need to clearly mention the biggest challenges in this field.

Ans: We appreciate your comment. In PCa management, no relevant clinical approaches are PCSCs based. In addition, PCSCs are inherently resistant to anti-androgen or chemotherapy treatment and they select to over-activate survival pathway in order to better adapt to the hostile tumor micro-environment. Such features make the eradication of PCSCs is very challenging. For this end, better understanding of PCSCs biology at the molecular level will definitely inspire us to develop PCSCs targeted approaches. You can reach this statement in the abstract, paragraph 55-60, paragraph128-133. Thanks.

Q3. Third, several references the authors cited in this study is too old. For review article, citing references older than 10 years is not recommended. Also, the style of the reference citing is not unified. I suggest the authors can base on the previous review (such as 10.3390/cancers11040434, and 10.3389/fonc.2022.935715) and update the latest finding in this review.

Ans: Your comments are highly suggestive. We updated and included the latest references related to this study in the revised review except the pioneer studies which were detailedly discussed in this review. Thanks

Q4. Forth, microRNAs and glucocorticoid receptor signaling are also play critical role in the stemness and androgen escape in PCa. I suggest the authors discuss them.

Ans: Thanks to your suggestion. The contribution of glucocorticoid receptor signaling to the stemness and androgen escape in PCa is fully discussed in paragraph 391-404. However, microRNAs and long non-coding RNAs were excluded from this review because they do not work independently and rely on their targeting genes to exert biological functions. We also include this rationale in paragraph 389-392. Thanks.  

Reviewer 2 Report

The topic taken up by the Authors is important for society because prostate cancer (PCa) is one of the mortal cancers. Their attention is focused on prostate cancer stem cells (PCSCs). In this review, they describe various signaling pathways that are responsible for the homeostatic regulation of PCSCs. Based on the current knowledge they discuss and provide some research perspectives on how PCSCs could be eliminated in clinical practice. The work is well-thought, well-organized, and includes most published articles. Generally, I have not found significant limitations in this manuscript; conversely, I think that it has many strengths, such as originality and accurate presentation of the issue.

I have a few minor remarks:

- In Table 1: There should be ‘CD117’ instead of ‘CD177’ (in the Function column).

- Explain the meaning of ‘PTEN’ or ‘CRPC’ at the first use, no later as it is. No explanation for ‘JAK’ or ‘STAT’. Generally, you use a lot of abbreviations so a list of them would be desirable.

- Pay attention to the notation of references – it should be consistent throughout the work (e.g., line 30: [11] [12,13] or [11-13], line 102: [47] [48] or [47,48]). The same for ‘et al’ or ‘et al’, ‘homodimer’ or ‘homo-dimer’, ‘WNT’ or ‘Wnt’.

- Figure 2 is difficult to read. Maybe you should somehow mark the signaling pathways mentioned in the text for better clarity.

- There are a lot of editing errors, mainly no or double spaces. Look also at lines 125 (interleukins) or 158 (androgenic hormones).

Author Response

Reviewer #2 (Remarks to the Author)

Q1. The topic taken up by the Authors is important for society because prostate cancer (PCa) is one of the mortal cancers. Their attention is focused on prostate cancer stem cells (PCSCs). In this review, they describe various signaling pathways that are responsible for the homeostatic regulation of PCSCs. Based on the current knowledge they discuss and provide some research perspectives on how PCSCs could be eliminated in clinical practice. The work is well-thought, well-organized, and includes most published articles. Generally, I have not found significant limitations in this manuscript; conversely, I think that it has many strengths, such as originality and accurate presentation of the issue.

Ans: Your positive comments are highly appreciated. Thanks

Q2. In Table 1: There should be ‘CD117’ instead of ‘CD177’ (in the Function column).

Ans: We appreciate your careful review. It has been corrected in the revised manuscript. Thanks

Q3. Explain the meaning of ‘PTEN’ or ‘CRPC’ at the first use, no later as it is. No explanation for ‘JAK’ or ‘STAT’. Generally, you use a lot of abbreviations so a list of them would be desirable.

Ans: We thank for your careful review. We went through the manuscript and explained the abbreviations in the revised manuscript. Also, the full name of each abbreviation was included in this review before the reference section. Thanks

Q4. Pay attention to the notation of references – it should be consistent throughout the work (e.g., line 30: [11] [12,13] or [11-13], line 102: [47] [48] or [47,48]). The same for ‘et al’ or ‘et al’, ‘homodimer’ or ‘homo-dimer’, ‘WNT’ or ‘Wnt’.

Ans: Again, your careful review is appreciated. We already made the relevant corrections. Thanks

Q5. Figure 2 is difficult to read. Maybe you should somehow mark the signaling pathways mentioned in the text for better clarity.

Ans: Thanks for the suggestion. The signaling pathways discussed in the review are marked in revised Figure 2. Thanks

Q6. There are a lot of editing errors, mainly no or double spaces. Look also at lines 125 (interleukins) or 158 (androgenic hormones).

Ans: We appreciate your careful review. The revised manuscript has been carefully edited by us and we believe this version would be improved a lot. Thanks